# Preliminary Structural Characterization of Selenium Nanoparticle Composites Modified by *Astragalus* Polysaccharide and the Cytotoxicity Mechanism on Liver Cancer Cells

**DOI:** 10.3390/molecules28041561

**Published:** 2023-02-06

**Authors:** Haiyu Ji, Xiaowei Lou, Jianshuang Jiao, Yang Li, Keyao Dai, Xiaoyu Jia

**Affiliations:** 1Center for Mitochondria and Healthy Aging, College of Life Sciences, Yantai University, Yantai 264005, China; 2College of Food Science and Engineering, Tianjin University of Science and Technology, Tianjin 300457, China; 3Tianjin Key Laboratory of Postharvest Physiology and Storage of Agricultural Products, Key Laboratory of Storage of Agricultural Products, National Engineering Technology Research Center for Preservation of Agricultural Products, Ministry of Agriculture and Rural Affairs, Tianjin 300384, China

**Keywords:** structural characterization, *Astragalus* polysaccharide-selenium nanoparticle composites, liver cancer cells, mitochondria pathway

## Abstract

*Astragalus* alcohol soluble polysaccharide (AASP) could present superior water solubility and antitumor activity with high concentration. Selenium nanoparticles (SeNPs) have received growing attention in various fields, but their unstable property increases the application difficulties. In the present study, functionalized nano-composites (AASP−SeNPs) were synthesized through SeNPs using AASP (average molecular weight of 2.1 × 10^3^ Da) as a surface modifier, and the preliminary structural characteristics and inhibitory mechanism on liver cancer (HepG2) cells were investigated. Results showed that AASP−SeNPs prepared under a sodium selenite/AASP mass ratio of 1/20 (*w*/*w*) were uniformly spherical with a mean grain size of 49.80 nm and exhibited superior dispersivity and stability in water solution. Moreover, the composites could dose-dependently inhibit HepG2 cell proliferation and induce apoptosis through effectively regulating mitochondria-relevant indicators including Δ*Ψm* depletion stimulation, intracellular ROS accumulation, Bax/Bcl-2 ratio improvement, and Cytochrome c liberation promotion. These results provide scientific references for future applications in functional food and drug industries.

## 1. Introduction

Hepatic carcinomas are the sixth most frequent malignant tumors globally, and the incidence and mortality are still increasing [1]. Although partial hepatic resection and liver transplantation have been successfully plied in treatments of the early-stage HCC, the follow-up therapeutic effects remain dissatisfactory because of the high probability of relapse after operation [2]. In addition, widely used chemotherapy also exhibited some disadvantages, including significant tolerance of cancer cells to drugs and non-targeted cytotoxic effects, leading to difficulties in HCC therapy [3]. Therefore, it is urgent to develop alternative treatments to reduce the risk of postoperative recurrence of HCC.

Nano-technology has been universally applied in food and pharmaceutical industries recently, which provides innovative thinking for the diagnosis and the treatment of some common diseases including tumors [4,5]. Many nanomaterials, such as nano-selenium, have successfully served as curative drugs or pharmaceutical carriers due to the underlying biological activities [6]. Selenium, one of the essential microelements for organisms, plays a vital role in preventing various illness, particularly in preventing malignancy and inducing tumor cell apoptosis [7,8]. Increasing evidence has revealed that nanoscale selenium has greater advantages due to the increased safety and hypotoxicity than other types of selenium [9]. However, unmodified nano-selenium is generally unstable and tends to aggregate, thereby greatly reducing its bioavailability [10]. Studies manifest that many biological macromolecules (such as polysaccharides, proteins, polyphenols, etc.) can serve as modifiers and regulators to coat nano-selenium, and have been proved to act as potential anticancer drug candidates [11,12,13]. Polysaccharides are considered the best choice for the synthesis of nano-selenium because of the abundant terminal hydroxyl and large specific area, easy preparation, good biocompatibility, high biological activity, and low toxicity [14,15,16].

In our previous research, *Astragalus* alcohol soluble polysaccharide (AASP, 2.1 × 10^3^ Da) was successfully isolated from the alcohol-soluble component of *Astragalus* water leaching solution and exhibited low toxicity and good water solubility, which also effectively inhibited the growth of H22 solid tumors in tumor-bearing mice [17]. Given the above distinctive advantages, it is reasonable to consider that AASP has the potential to be a natural template synthesizing stable nano-selenium, which may give full play to its superior biological characteristics. Therefore, this paper aimed to prepare AASP-SeNP composites using AASP as a modifier, and to research the preliminary structural characterization and molecular mechanism of anti-hepatoma activity, which would be of great theoretical significance for further applications in food and medicine fields.

## 2. Results

### 2.1. Synthesis of AASP−SeNPs

In the present study, AASP-SeNP composites were prepared using AASP as the stabilizer and dispersant in a redox system containing Na_2_SeO_3_ and ascorbic acid, and the morphological and structural features are shown in Figure 1.

Figure 1A displays photos of AASP-SeNP aqueous solutions with five diverse mass ratios of Na_2_SeO_3_ to AASP and the SeNP solution decorated without AASP (as a control) at 0 and 35 days. The freshly prepared AASP-SeNP solutions changed gradually from slightly cloudy orange-red to transparent orange with Na_2_SeO_3_/AASP ratios (*w*/*w*) increasing from 1:10 to 1:20 and finally faded to pale yellow with increasing ratios from 1:20 to 1:30, indicating that the Na_2_SeO_3_/AASP ratios can effectively control the performance of SeNPs, and excess addition of AASP in a redox system might not be conducive to the formation of SeNPs, which was similar to previous research [15,18]. Additionally, undecorated SeNP solution was cloudy, and the SeNPs gathered and settled in aqueous solution after 35 days, perhaps due to its high surface energy [19]. By contrast, AASP−SeNPs exhibited a good dispersibility and superior stability on the 35th day, except for the Na_2_SeO_3_/AASP ratios of 1:10. A small amount of precipitate was observed in the bottom when the Na_2_SeO_3_/AASP ratio was 1:15. The phenomenon indicated that an appropriate amount of AASP might be helpful for the synthesis and stability of SeNPs, which could result from abundant hydroxyl groups in AASP molecules and the electrostatic repulsion between molecules.

As displayed in Figure 1B, the mean grain sizes of AASP-SeNP composites declined dramatically from 166.89 nm to 49.80 nm when the mass ratios of Na_2_SeO_3_ to AASP increased from 1:10 to 1:20 and increased remarkably from 49.80 nm to 96.67 nm with increasing ratios from 1:20 to 1:30. As reported, SeNPs with smaller particle sizes were more stable and possessed more significant inhibitory effects on the growth of cancer cells [20]. As shown, the stability of nanoparticles was of fundamental importance for assessing their practical applications. Zeta potential is a crucial indicator for stability assessment. Figure 1C shows that all AASP-SeNP composites at various Na_2_SeO_3_/AASP ratios exhibited negative zeta potentials in aqueous solution. When the Na_2_SeO_3_/AASP ratio was 1:20, the prepared AASP−SeNPs exhibited the maximal negative zeta potential (−37.08 mV) in comparison with other groups, implying the best stability of the nano-selenium (Na_2_SeO_3_/AASP = 1:20). These taken together, the Na_2_SeO_3_/AASP ratio of 1:20 was chosen for follow-up experiments.

### 2.2. Characterization and Synthetic Mechanism of AASP−SeNPs

The morphology and elemental distribution of AASP−SeNPs were analyzed using TEM and STEM analysis, and these findings are presented in Figure 2A–F. The SeNPs coated with AASP displayed evenly distributed spherical particles and contained mainly Se, C, and O elements. The Se atomic content reached 11.21%, confirming that AASP played essential roles in the dispersibility of SeNPs.

Figure 2G presents the UV spectra (200–800 nm) of AASP and AASP−SeNPs. There were no significant peaks at 260 nm and 280 nm in the UV–vis spectrum of AASP, demonstrating that AASP contained little or no nucleic acids and proteins. In contrast with AASP, AASP−SeNPs exhibited large characteristic peaks at 260~270 nm, which may have resulted from the surface plasmon resonance of AASP-SeNP composites, further suggesting the existence of an interaction between AASP and SeNPs [21].

As shown in Figure 2H, the reciprocal action between SeNPs and AASP was further verified by comparing FTIR spectra of AASP and AASP-SeNP composites. In the spectrum of AASP, three typical absorption peaks at 3430, 2922, and 1631 cm^−1^ were attributable to −OH, C−H bending vibration, and bound water, respectively [22]. Three characteristic peaks around 1050 cm^−1^ were ascribed to the presence of a pyranose ring in AASP. No new peaks were reflected in the FTIR spectrum of AASP−SeNPs, indicating that no new covalent bonds were produced in the functionalized nano-selenium. However, a remarkable shift from 3430 cm^−1^ (AASP) to 3410 cm^−1^ (AASP−SeNPs) was noticed, confirming the mutual effect between hydroxyl of AASP and Se atoms of SeNPs, which might result in the spherical morphology and superior stability of nano-selenium [23,24].

### 2.3. Inhibitory Effects of AASP−SeNPs on HepG2 Cells

The MTT results revealed the suppressive effects of AASP−SeNPs on human hepatocellular carcinoma HepG2 cells with increasing drug concentrations and prolonged incubation time. As seen from Figure 3, the inhibition of AASP−SeNPs on HepG2 cells was greatly increased with incremental dosages ranging from 25 to 800 μg/mL, showing the concentration correlativity. Additionally, the inhibitory rates were higher under the same concentration of AASP−SeNPs for 48 h than those for 24 h, verifying that AASP−SeNPs could restrain HepG2 cell growth with a temporal correlation, and the IC_50_ values were 177.79 μg/mL for 24 h, 77.48 μg/mL for 48 h. Considering the short-term efficient treatment, three concentrations (100, 200, and 400 μg/mL) with an exposure time of 24 h were chosen for the following experiments.

### 2.4. Morphological Alterations and Apoptotic Rates of HepG2 Cells

Morphological alterations of AASP-SeNP-treated HepG2 cells were visualized under an inverted fluorescence microscope. As presented in Figure 4A, normal HepG2 cells exhibited an intact shape and good anchorage dependence. In contrast, AASP-SeNP treatment remarkably reduced the cell number with increasing concentrations and caused some typical morphological changes including prominent membrane shrinkage and incomplete attachment to the culture flask. In addition, seen from the Hoechst 33258 staining results (Figure 4B), the cell nuclei of untreated HepG2 cells were regular, large, and dyed uniformly, whereas the AASP-SeNP-treated cells presented significant pyknosis and DNA fragmentation.

The apoptotic rates induced by AASP−SeNPs on HepG2 cells were further evaluated via the Annexin V-FITC/PI double staining method. Seen from Figure 4C,D, AASP-SeNP treatment effectively increased both the early and late apoptotic cell ratios in comparison with the normal HepG2 cells (*p* < 0.05), which exhibited a dose-dependent response. The total apoptotic rates of HepG2 cells increased from 3.03 ± 0.20% to 27.02 ± 0.67%, 39.22 ± 2.22%, and 55.43 ± 2.51%, respectively, with AASP-SeNP concentrations increased from 0 to 100, 200, and 400 μg/mL. The apoptosis-induced effect was one of the crucial events for selenium or selenium compounds to perform cancer prevention and treatment [25]. Some studies indicated that a modifying agent improved the permeability and targeted intracellular uptake of nano-selenium [26], which might be responsible for the intensive antitumor efficacy of AASP−SeNPs.

### 2.5. Effects of AASP−SeNPs on Cell Cycle Distributions of HepG2 Cells

The cell cycle is a well-organized progress that can usually be split into G1, S, G2, and M phases. Particularly, uncontrollable or disordered cell cycles can result in tumorigenesis and cell apoptosis, which are important indicators for novel antineoplastic drug development [27]. Thus, the influence of AASP−SeNPs on cell cycle distributions was detected employing flow cytometry. As indicated in Figure 5, AASP−SeNPs significantly downregulated the percentages in G0/G1 phase in comparison with the untreated HepG2 cells (*p* < 0.05), resulting in the increase of cell numbers in S phase (*p* < 0.05), which suggests that AASP−SeNPs might induce apoptosis of HepG2 cells by triggering S-phase arrest.

### 2.6. ROS and *Δ*Ψm Levels in Mitochondria of HepG2 Cells

Mitochondria, famous as important intracellular organelles for producing energy, are considered a regulatory center of both cell survival and death [28]. The levels of Δ*Ψm* are an important indicator of mitochondrial states, and a decline in Δ*Ψm* is known as an early warning signal for cell apoptosis [21]. The Δ*Ψm* of HepG2 cells treated with or without AASP−SeNPs was investigated via JC-1 staining and flow cytometric analysis. Seen from Figure 6A,B, the ratios of red to green fluorescence of JC-1 reduced notably from 11.03 ± 0.18 to 7.30 ± 0.22, 5.51 ± 0.49, and 1.56 ± 0.13 with increasing concentrations of AASP−SeNPs from 0 to 100, 200, and 400 μg/mL, implying that the depletion of Δ*Ψm* induced by AASP−SeNPs was responsible for the apoptosis of HepG2 cells.

Furthermore, ROS expression in HepG2 cells was evaluated using DCFH-DA analysis, and the variations in DCF fluorescence intensity are displayed in Figure 6C,D. A dose-dependent enhancement of fluorescence intensity was observed in HepG2 cells after treatment with increasing dosages of AASP−SeNPs for 24 h, compared to untreated cells (*p* < 0.05). Accumulating evidence has demonstrated that ROS are the secondary products of cellular metabolism which are crucial for normal physiological processes; however, the excessive accumulation of ROS can lead to oxidative stress and can then trigger the apoptosis of cancer cells [29,30]. The results showed that intracellular ROS accumulation participated in the progress of AASP-SeNP-induced HepG2 cell apoptosis.

### 2.7. Apoptosis-Associated Protein Levels of HepG2 Cells

AASP-SeNP-regulated mitochondrial molecular mechanisms were detected using Western blot analysis. As shown in Figure 7A–C, AASP−SeNPs elevated Bax expression and decreased Bcl-2 expression, dose-dependently. Increasing ratios of Bax/Bcl-2 proteins caused variations in mitochondrial membrane permeability and further induced the liberation of Cytochrome c from mitochondria to cytoplasm in HepG2 cells with the dosages of AASP−SeNPs elevating from 0 to 400 μg/mL, which was a crucial course of cell apoptosis [31,32].

## 3. Materials and Methods

### 3.1. Materials and Reagents

*Astragalus* roots were purchased from Tianjin Tongrentang Group Co., Ltd. (Tianjin, China). Bax, β-actin, Bcl-2, and Cyt c antibodies were purchased from ImmunoWay Biotechnology Company (Plano, TX, USA). Apoptosis index-related detection kits for cell experiments were all provided by Beijing Solarbio Science & Technology Co., Ltd. (Beijing, China). Human hepatoma HepG2 cells were provided by the Shanghai Institute of Biological Sciences at the Chinese Academy of Sciences (Shanghai, China). All other chemicals used in this study were of analytical grade.

### 3.2. Synthesis of AASP−SeNPs

AASP-SeNP composites were synthesized following the previously reported method with slight modification [33]. Briefly, AASP powder (60 mg) was distilled in double-distilled water (30 mL) and mixed with different volumes of freshly prepared Na_2_SeO_3_ solution (0.01 M), with Na_2_SeO_3_ to AASP ratios of 1:10, 1:15, 1:20, 1:25, and 1:30 (*w*/*w*), respectively, under magnetic agitation. Afterwards, the same volumes of vitamin c solution (0.04 M) were mixed with AASP–Na_2_SeO_3_ solution drop by drop, stirring continually in the dark for 4 h under greenhouse conditions. The reaction solution was dialyzed (molecular weight cut-off of 1000 Da) against distilled water and freeze-dried for further research. Moreover, SeNPs without polysaccharide were obtained according to the above approach by replacing AASP aqueous solution with the same volume of double-distilled water.

### 3.3. Physicochemical Properties of AASP−SeNPs

The prepared AASP−SeNPs were identified by the following methods. In brief, the mean particle diameter and size distribution of AASP−SeNPs were evaluated via a laser nanoparticle size analyzer (BT-90, Baite Co., Ltd., Dandong, China) at 25 ± 0.1 °C. The zeta potentials were detected by a zeta potential analyzer (Szp-06, BTG-Mutek, Herrsching, Germany). The morphology of AASP−SeNPs was visualized under a transmission electron microscope. Moreover, the elemental composition of AASP−SeNPs was identified using mapping analysis. The UV spectra of AASP and the nano-selenium composites were determined under a microplate reader. The characteristic functional groups of AASP and the composites were evaluated via a Bruker VECTOR-22 Fourier Transform Infrared Spectroscopy analysis.

### 3.4. Anti-Hepatoma Activity of AASP−SeNPs In Vitro

#### 3.4.1. MTT Assay

The inhibitory effects of AASP−SeNPs on HepG2 cells were evaluated through an MTT assay [34]. Initially, 1 × 10^4^ cells/well were inoculated into a 96-well plate and cultured in an incubator (37 °C, 5% CO_2_) overnight. AASP-SeNP solution with different final concentrations (0–800 μg/mL) was seeded for 24 h, followed by an addition of MTT for 4 h. Subsequently, the culture solution was discarded, and then 150 µL DMSO was put into each well to dissolve the reaction product adequately. The optical density (OD) was detected at 570 nm. The inhibition ratio (IR) was calculated to assess the antitumor capacities of AASP−SeNPs on HepG2 cells, and the calculation formula is as follows: IR (%) = (OD_untreated cells_ − OD_AASP−SeNPs-treated cells_)/OD_untreated cells_ × 100.

#### 3.4.2. Cell Morphological Observation

After the HepG2 cells were treated with AASP−SeNPs (0–400 μg/mL) for 24 h, the morphological alterations of HepG2 cells (with/without Hoechst 33258 staining) were observed by an inverted fluorescence microscope (Nikon, Tokyo, Japan).

#### 3.4.3. Cell Apoptosis Assay

HepG2 cells were exposed to AASP−SeNPs (0–400 μg/mL) for 24 h, then treated by Annexin V-FITC and PI double staining. In brief, the treated cells were collected, washed with PBS solution three times, and resuspended in 1 mL 1× binding buffer. Thereafter, the cells were stained according to the manufacturer’s instruction. The apoptotic rates were detected and analyzed by flow cytometry.

#### 3.4.4. Cell Cycle Distribution

The distributions and proportions of the cell cycle in HepG2 cells after AASP-SeNP treatments were investigated using a DNA content quantitation assay. Cells treated by AASP−SeNPs (0–400 μg/mL) for 24 h were obtained, washed with PBS solution three times, followed by fixation with 70% pre-cooled ethanol solutions for 4 h. The fixed cells were stained in accordance with the instructions of the test kits. The cell cycle distributions were tested and analyzed by flow cytometry.

#### 3.4.5. Mitochondrial Membrane Potential (Δ*Ψm)* Determination

The Δ*Ψm* of HepG2 cells after AASP-SeNP treatments was investigated with JC-1 staining. After exposure to AASP−SeNPs (0–400 μg/mL) for 24 h, the cells were obtained, washed with PBS solution three times, and incubated with JC-1 staining solutions according to the instructions, and then detected by flow cytometry.

#### 3.4.6. ROS Assay

The intracellular ROS changes of HepG2 cells were detected by a reactive oxygen species assay kit. Briefly, the collected cells treated with AASP−SeNPs (0–400 μg/mL) for 24 h were washed with PBS solution three times and incubated with DCFH-DA solution (10 μmol/L). After being washed again, these stained cells were examined and analyzed by flow cytometry.

#### 3.4.7. Apoptosis-Related Protein Detection

The expressions of apoptotic proteins in AASP-SeNP-treated HepG2 cells were determined through the Western blotting method. The total proteins were extracted from HepG2 cells and separated using 12% SDS-PAGE, followed by an electrotransfer into a PVDF membrane. Bax, Bcl-2, Cyt c, and β-actin expression levels were detected by using corresponding antibodies. Ultimately, the specific bands were visualized in X-ray films in the dark via chemiluminescent immunoassay. The gray values of related proteins were analyzed by Image J software.

### 3.5. Statistical Analysis

All values obtained in this study are expressed as mean ± SD (standard deviation). The between-group differences were evaluated by Student’s t-test and one-way analysis of variance (ANOVA), and the between-group significance was defined at *p* < 0.05.

## 4. Discussion

Stable and size-controlled nano-selenium was feasibly synthesized by an eco-friendly green and low-cost approach via ascorbic acid as the reductant and natural polysaccharide as a stabilizing agent In our previous research, the AASP was isolated from *Astragalus* roots with a relatively low formula weight of approximately 2100 Da and exhibited remarkable antineoplastic activities on H22-bearing mice by regulating their body immunity. AASP is a very easily neglected component in the traditional polysaccharide extraction process, and its further development and utilization will effectively improve the utilization rate of *Astragalus membranaceus*. Its superior biological activity will also promote development in various fields. Therefore, considering that SeNPs are easy to aggregate and AASP possesses many reactive hydroxyl groups, it can be inferred that AASP has the potential to be involved in the nucleation, formation, and stability of SeNPs [15]. Particle size is an important element that affects the physicochemical and biological characteristics of nanomaterials [35]. Changed mass ratios of Na_2_SeO_3_ to polysaccharides can influence the particle size of nano-selenium [36]. Therefore, the effect of different Na_2_SeO_3_/AASP ratios (*w*/*w*) on the average sizes of AASP-SeNP composites was investigated. Moreover, the morphology and elemental distribution of the AASP−SeNPs were analyzed using TEM and STEM analysis. The results demonstrated that AASP-SeNP composites presented a monodisperse orbicular structure and high stability with a mean grain size of 51.82 nm in water, which indicated that the composites have the potential to exert superior antitumor biological activity.

Based on the successful preparation of stable nano-composites, we proceeded to study the inhibitory effects on HepG2 cells in vitro via an MTT test, morphological alterations, Annexin V-FITC/PI double staining method, cell cycle determination, ROS, and Δ*Ψm* level detection and obtained significant experimental results. As reported, the apoptotic cells were usually accompanied by some characteristic morphological features such as cellular membrane changes, shrinkage, nuclear concentration, and fragmentation [37,38]. The results demonstrated that AASP−SeNPs could effectively exert cytotoxicity and induce apoptosis in HepG2 cells, dose-dependently. Additionally, in recent years, a large number of studies have reported that apoptosis can also lead to cell cycle disorder, decreased mitochondrial transmembrane potential, and up-regulation of intracellular reactive oxygen species levels [39,40]. As displayed, the typical apoptotic indicators were all detected in the AASP-SeNP-treated HepG2 cells, suggesting effective apoptosis induction effects of these nano-composites.

In the Bcl-2 family, the changing contents of Bax and Bcl-2 proteins exert a crucial effect on mitochondria-dependent apoptosis cellular signal transduction [41,42]. As is known, the liberation of Cytochrome c can activate caspase-9 and further cleave caspase-3, which causes systematic vandalization during cell apoptosis [43,44]. Therefore, Cytochrome c plays a crucial role in the caspase-mediated mitochondrial pathway. In this paper, AASP−SeNPs remarkably elevated Bax expression while decreasing Bcl-2 expression and further increased Cytochrome c liberation, indicating the disruption of mitochondrial membranes and induction of apoptosis, which was consistent with previous results.

## 5. Conclusions

In conclusion, the obtained AASP-SeNP composites exhibited a monodisperse orbicular structure and high stability with a mean grain size of 49.80 nm in water, when the Na_2_SeO_3_/AASP ratio (*w*/*w*) was 1:20. Additionally, AASP−SeNPs exhibited remarkable cytotoxicity toward HepG2 cells dose-dependently, accompanied by a series of typical apoptotic phenomena. Furthermore, AASP−SeNPs effectively activated the ROS- and caspase-mediated mitochondrial pathway by stimulating the depletion of Δ*Ψm*, inducing the accumulation of intracellular ROS, increasing Bax/Bcl-2 ratios, and promoting Cyt c liberation, finally leading to HepG2 cell apoptosis. In summary, AASP−SeNPs exhibited significant inhibitory effects on HepG2 cells by inducing ROS- and mitochondria- mediated apoptosis.

## Figures and Tables

**Figure 1 molecules-28-01561-f001:**
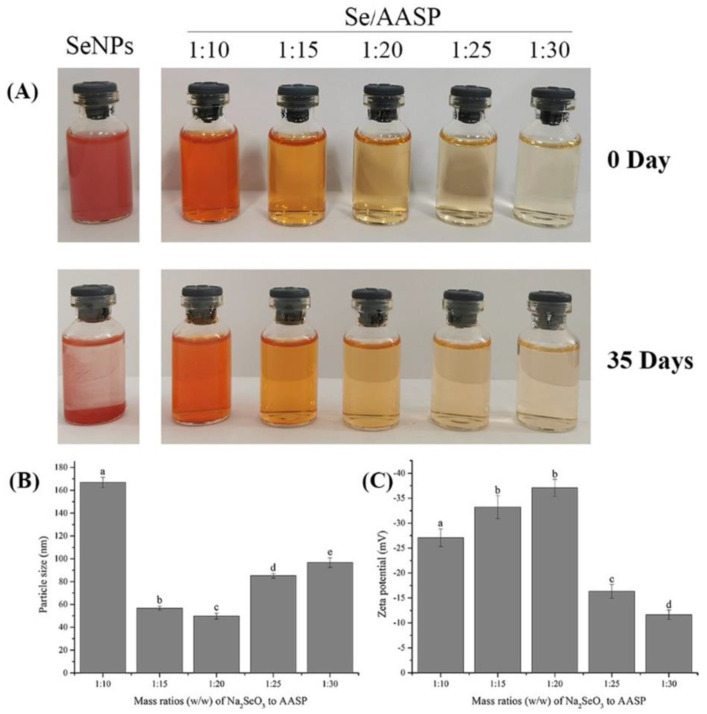
Preparation and characteristics of AASP−SeNPs. Photographs of AASP-SeNP solution prepared with different Na_2_SeO_3_ to AASP mass ratios at 0 and 35 days at 4 °C, respectively (**A**). Influences of different Na_2_SeO_3_ to AASP mass ratios on particle sizes (**B**) and zeta potentials (**C**). Different letters: a, b, c, d, e indicate that samples are significantly different.

**Figure 2 molecules-28-01561-f002:**
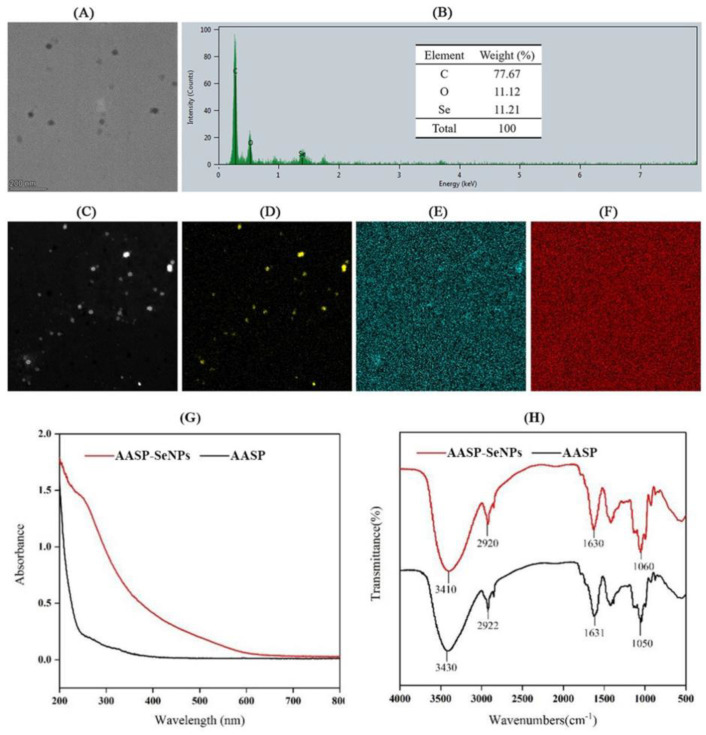
TEM (**A**), EDX spectrum (**B**), STEM (**C**), and STEM-EDX mapping images of Se (**D**), C (**E**), and O (**F**) elements of AASP−SeNPs. UV–vis absorption (**G**) and FTIR spectra (**H**) of AASP and AASP−SeNPs.

**Figure 3 molecules-28-01561-f003:**
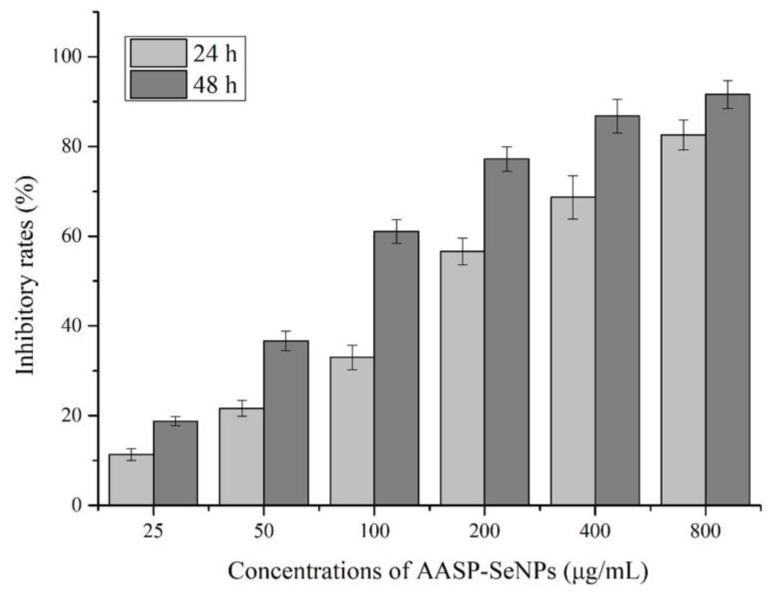
Inhibitory effects of different concentrations of AASP−SeNPs on HepG2 cells.

**Figure 4 molecules-28-01561-f004:**
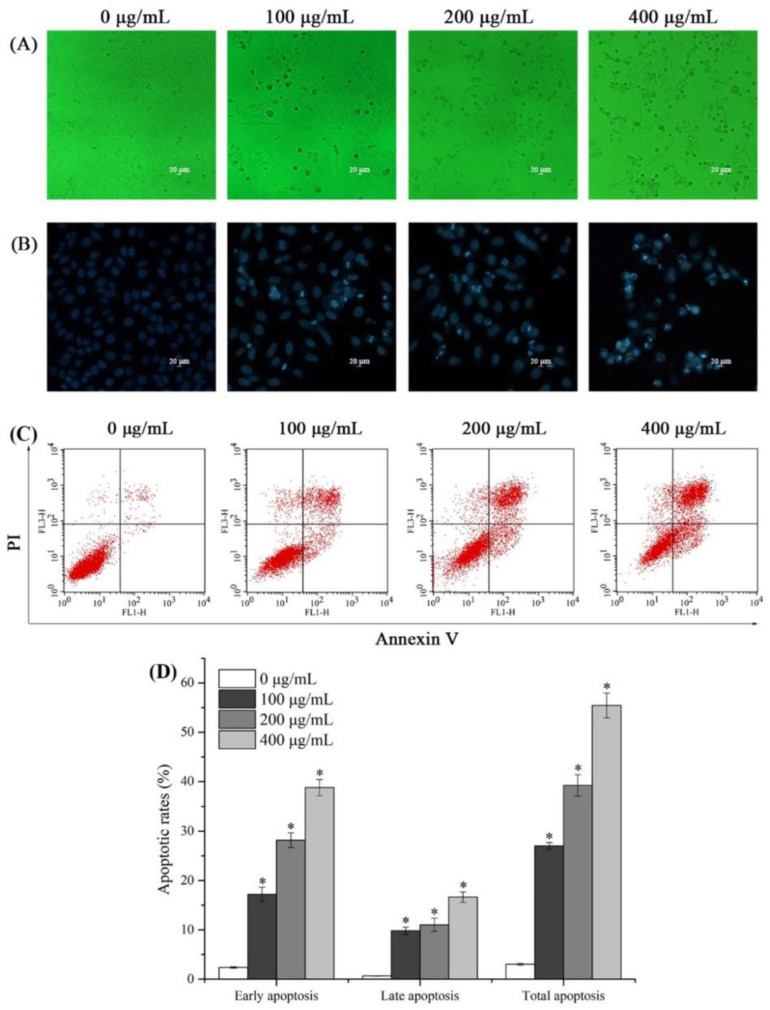
Optical photographs (**A**), 10×, fluorescence pictures (**B**), 10×, Annexin V-FITC and PI double staining distributions (**C**), and analysis (**D**) of AASP-SeNP-treated HepG2 cells. Note: *, *p* < 0.05, compared with normal HepG2 cells.

**Figure 5 molecules-28-01561-f005:**
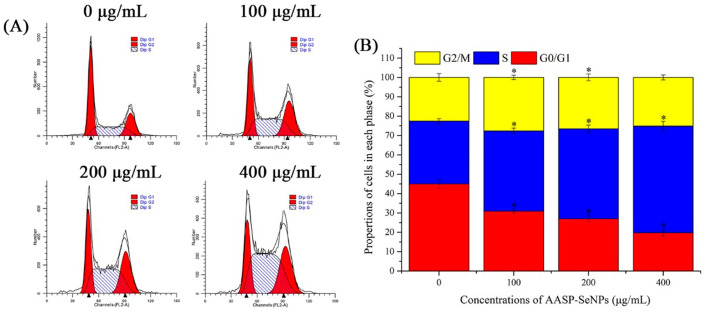
Cell cycle distributions (**A**) and corresponding percentages (**B**) of AASP-SeNP-processed HepG2 cells. Note: *, *p* < 0.05, compared with normal HepG2 cells.

**Figure 6 molecules-28-01561-f006:**
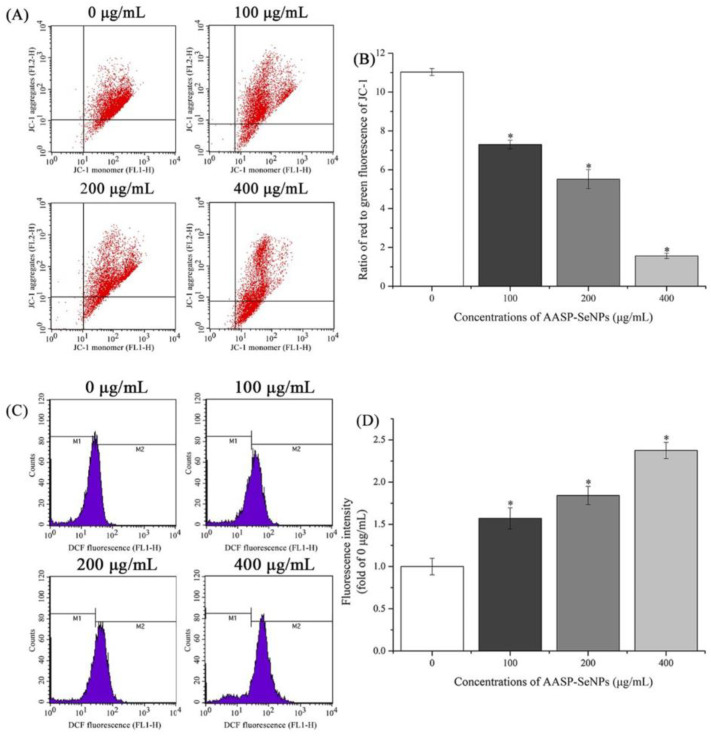
ROS-mediated mitochondrial state detection of AASP-SeNP-processed HepG2 cells exposed for 24 h. Scatter diagram (**A**) and column diagram (**B**) reflecting the mitochondrial conditions of AASP-SeNP-treated HepG2 cells via JC-1 staining using flow cytometry. Distributions (**C**) and proportions (**D**) showing the intracellular ROS accumulations of HepG2 cells. Note: *, *p* < 0.05, compared with normal HepG2 cells.

**Figure 7 molecules-28-01561-f007:**
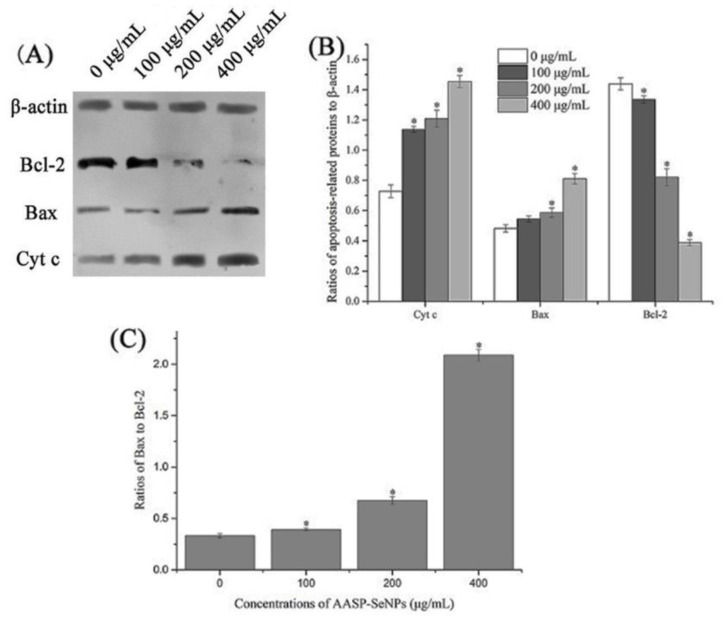
Apoptotic protein levels of AASP-SeNP-processed HepG2 cells. Western blot results of Cyt c, Bax, and Bcl-2 protein levels (**A**). Quantitative analysis for the levels of Cyt c, Bax, and Bcl-2 proteins relative to β-actin (**B**). Effect of AASP−SeNPs on Bax/Bcl-2 ratios (**C**). Note: *, *p* < 0.05 compared with normal HepG2 cells.

## Data Availability

Not applicable.

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
