# Peer review of "Preliminary Structural Characterization of Selenium Nanoparticle Composites Modified by Astragalus Polysaccharide and the Cytotoxicity Mechanism on Liver Cancer Cells"

_molecules, 2023, doi:10.3390/molecules28041561_

Round 1

Reviewer 1 Report

The presented article "Preliminary structural characterization analysis of Astragalus polysaccharide - selenium nanoparticles complex and the cytotoxicity mechanism on liver cancer cells" concerns an important topic for medicine and humanity. In my opinion, it can be published as is.  I have found a word in line 45 "rozpoznanie" that is not in English but in Polish and should be changed. 

Author Response

Reviewer 1:

Comments and Suggestions for Authors

The presented article "Preliminary structural characterization analysis of Astragalus polysaccharide - selenium nanoparticles complex and the cytotoxicity mechanism on liver cancer cells" concerns an important topic for medicine and humanity. In my opinion, it can be published as is.  I have found a word in line 45 "rozpoznanie" that is not in English but in Polish and should be changed.

Response: Thank you very much for the kind suggestion. The word in line 45 "rozpoznanie" has been revised. Besides, the English writing of this manuscript has been further carefully modified. We hope that the qualities of revised version could be obviously improved.

Reviewer 2 Report

The manuscript is well formulated and presents a topic of interest by presenting the phytosynthesis of selenium nanoparticles and their antitumor activity. A single observation for the authors, references 38 and 39 were placed after reference 40.

I believe that the manuscript meets the conditions to be published, after correcting the references.

Author Response

Reviewer 2:

Comments and Suggestions for Authors

The manuscript is well formulated and presents a topic of interest by presenting the phytosynthesis of selenium nanoparticles and their antitumor activity. A single observation for the authors, references 38 and 39 were placed after reference 40.

I believe that the manuscript meets the conditions to be published, after correcting the references.

Response: Thank you very much for the kind advice. The order of references 38, 39, 40 has been corrected, and the English writing level has been also further improved. We hope that the qualities of revised manuscript could be enhanced and meet the publication requirements.

Reviewer 3 Report

1.      Change the title.

2.      In title, author mention about the polysaccharide. But no details about the polysaccharide. How the polysaccharides were isolated?

3.      Line n80: check this line "AASP-SeNPs composites"

4.      the characterization part is not enough, add more structural confirmations, Add more TEM images with good magnification

5.      Include the toxicity assessment for AASP-SeNPs composites

6.      Add better UV analysis, image. there is no peak shown for uv analysis

7.      Include XRD analysis results both AASP and AASP-SeNPs

Author Response

Reviewer 3:

Comments and Suggestions for Authors

  1. Change the title.

Response: Thank you very much for the valuable suggestion. The tittle of “Preliminary structural characterization analysis of Astragalus polysaccharide - selenium nanoparticles complex and the cytotoxicity mechanism on liver cancer cells” has been changed to “Preliminary structural characterization of selenium nanoparticles composites modified by Astragalus polysaccharide and the cytotoxicity mechanism on liver cancer cells”.

  1. In title, author mention about the polysaccharide. But no details about the polysaccharide. How the polysaccharides were isolated?

Response: Thanks very much for the kind advice. The Preliminary structural characterization analysis (UV, FTIR, GC) of Astragalus polysaccharide has been published in our earlier paper “Yu, J.; Ji, H.-y.; Liu, A.-j. Alcohol-soluble polysaccharide from Astragalus membranaceus: Preparation, characteristics and antitumor activity. International Journal of Biological Macromolecules 2018, 118, 2057-2064, doi:10.1016/j.ijbiomac.2018.07.073”, which has been cited in reference 17. However, we further detect the specific structural characteristics of this polysaccharide, and the relevant data would be submitted in our future manuscript. In this paper, we aimed to investigate the preliminary structural characterization of selenium nanoparticles composites modified by Astragalus polysaccharide and the cytotoxicity mechanism on liver cancer cells, the polysaccharide was used as a decorating material, therefore the details about the polysaccharide were not presented in this manuscript, we hope you could understand.

  1. Line n80: check this line "AASP-SeNPs composites"

Response: Thank you very much for the valuable advice. The expression of sentence in line 80 has been improved.

  1. the characterization part is not enough, add more structural confirmations, Add more TEM images with good magnification

Response: Thank you very much for the professional advice. We agree and appreciate your opinion very much, the characterization part of polysaccharide and the composites is inadequate, which has been indicated in the tittle “Preliminary structural characterization of selenium nanoparticles composites modified by Astragalus polysaccharide”, and that is why we put a lot of efforts into detecting apoptotic indicators of HepG2 cells. Besides, some reasons have been involved in response 2, the TEM images were used for analyze elements contents of the complex in a semi-quantitative manner, therefore we consider these presented TEM images could meet the demands of this paper.

  1. Include the toxicity assessment for AASP-SeNPs composites

Response: Thank you very much for the kind advice. The toxicity assessment for AASP-SeNPs composites has been provided in section 3.4 “Growth inhibition effect of AASP-SeNPs on HepG2 cells”, and the related data was displayed in Fig. 3 Inhibitory effects of different concentrations of AASP-SeNPs on HepG2 cells. These results were used for dosage selection for later experiments.

  1. Add better UV analysis, image. there is no peak shown for uv analysis.

Response: Thanks for this valuable suggestion. The relevant expressions have been modified. For UV spectrum of AASP, no obvious peak shown is reasonable, as presented in the main text, the result suggested that AASP contained little or no nucleic acids and proteins. For UV spectrum of AASP-SeNPs, the huge characteristic peak of 260~270 nm indicate the surface plasmon resonance of AASP-SeNPs composites, and the high content of the groups that produce the signal peak results in a wide peak rather than a uniformly distributed peak. Similar results are also published by other authors, such as “Xuelian Wang, Wenhui Liu, Yeling Li, Lingling Ma, Zhen Lin, Jing Xu, Yuanqiang Guo, Preparation and anti-tumor activity of selenium nanoparticles based on a polysaccharide from Paeonia lactiflora, International Journal of Biological Macromolecules, Volume 232, 2023, 123261, ISSN 0141-8130, https://doi.org/10.1016/j.ijbiomac.2023.123261.”.

  1. Include XRD analysis results both AASP and AASP-SeNPs.

Response: Thank you very much for the kind advice. X-Ray Diffraction technology has become the most basic and important means of structural testing, and its main applications are phases analysis and crystallinity determination. However, these indicators could not demonstrate significant significance in helping analyze in vitro antitumor activity of the composites in this manuscript, and polysaccharide and the nano-selenium composites in this paper have no crystalline morphology, so we did not conduct XRD detection like other authors investigating relevant studies.

Finally, as you mentioned “Extensive editing of English language and style required”, we further improved the expressions throughout this manuscript under the instruction of an experienced professor in our research team. We sincerely hope that this revised version could meet your and “molecules” journal’s requirements, thanks again for the review and these valuable comments.

Round 2

Reviewer 3 Report

Accept in present form